# Relationship between Self-Rated Health and Lifestyle and Food Habits in Japanese High School Students

**DOI:** 10.3390/bs7040071

**Published:** 2017-10-18

**Authors:** Tomoko Osera, Mitsuyo Awai, Misako Kobayashi, Setsuko Tsutie, Nobutaka Kurihara

**Affiliations:** 1Hygiene and Preventive Medicine, Graduate School of Home Economics, Kobe Women’s University, 2-1 Higashisuma-Aoyama, Suma, Kobe 654-8585, Japan; tomokocup12@yahoo.co.jp; 2Takakuradai Kindergarten attached to Kobe Women’s University, 4-2 Takakuradai, Suma, Kobe 654-0081, Japan; mkobayashi@suma.kobe-wu.ac.jp; 3School of Nursing, Kansai University of Nursing and Health Science, 1456-4 Shizuki, Awaji-City 656-2131, Japan; m.awai@kki.ac.jp; 4Clinical Nutrition Management, Graduate School of Home Economics, Kobe Women’s University, 2-1 Higashisuma-Aoyama, Suma, Kobe 654-8585, Japan; tsuti@suma.kobe-wu.ac.jp

**Keywords:** self-rated health (SRH), adolescence, cross-sectional study

## Abstract

Self-rated health (SRH), a subjective assessment of health status, is extensively used in the field of public health. It is an important and valid measure that is strongly related to morbidity, mortality, longevity and health status. Adolescence is a crucial period for the formation of health status, because health-risk behaviours (e.g., skipping breakfast) are often established during this period. In this study, we investigated the relationship of SRH with lifestyle and eating habits in Japanese high school students. In this study, 1296 students aged 16–18 years from 11 high schools in Japan participated. A questionnaire was administered to these participants that included a question on SRH, five questions on demographic characteristics, six questions on lifestyle items (e.g., wake-up time), five questions on miscellaneous health issues (e.g., anorexia), and 25 questions on food habits and attitudes towards food. We examined the differences between self-rated healthy and unhealthy groups using logistic regression analysis adjusted for gender and age. A dichotomy regression analysis was performed using a stepwise elimination method. Of the 1296 respondents, 16.7% reported feeling unhealthy, 57.7% of whom were females. The self-rated healthy group had a higher frequency of eating breakfast (odds ratio (OR): 2.13; confidence interval (CI): 1.07–4.24) and liked home meals to a greater extent (OR: 3.12; CI: 1.27–7.65) than the self-rated unhealthy group. The two groups did not differ significantly in terms of other lifestyle factors or unidentified complaints. Our results suggest that liking home meals during adolescence may lead to the development of good eating habits, i.e., eating breakfast, and better SRH.

## 1. Introduction

Self-rated health (SRH), a subjective assessment of health status, is extensively used in the field of public health. The validity of SRH measures, in terms of their ability to predict mortality risk, has been established in multiple studies [1]. The utility of SRH assessments has been shown in people with moderate-to-severe chronic kidney disease [2]. It is an important and valid measure, strongly related to morbidity, mortality, longevity and health status [1,2]. In addition, SRH is affected by several lifestyle, diet and biochemical factors [3,4].

Although SRH is a useful tool for estimating the subjective health needs of populations, some research has revealed that SRH may capture different aspects of the health of different social groups [5,6,7]. For example, blacks and whites may have different needs for health care compared to other population groups [8]. In addition, different profiles of social and behavioural determinants of SRH among patients with diabetes have been shown across countries [9]. Thus, local data is needed because of the considerable cross-racial, cross-ethnic, and cross-country variations in meanings and correlates of SRH.

Adolescence is a crucial period for the formation of health status [10,11,12]. Indeed, many health-related behaviours are formed during this period, including smoking, alcohol consumption, drug use, nutrition-related behaviours and physical activity [13]. During adolescence, health is also strongly affected by social factors, including income inequality, family support, school environment and peer influence [14].

SRH assessment among adolescents is a good measure of evaluating health-rated self-concepts. Specifically, SRH is strongly associated with general well-being and psychosomatic symptoms [11,15,16,17]. Research has suggested that SRH is influenced by gender, family support, lifestyle and psychological factors [13]. Moreover, one study has demonstrated a significant positive association between SRH and self-rated eating habits, suggesting that dietary habits influence overall perceived health [18]. However, there is a paucity of studies on SRH during adolescence. In addition, no study has shown a relationship between SRH and food habits of adolescents. Therefore, in this study, we investigated the association of SRH with lifestyle and eating habits in Japanese high school students.

## 2. Materials and Methods

### 2.1. Design and Data Collection

A cross-sectional study was conducted from May to November, 2015. In total, 1300 students aged 16–18 years from 11 high schools in Hyogo and Mie, Japan, participated in the study. The study areas were rural and urban. Of the total participants, 1296 (99.6%) returned the completed questionnaire, and all provided informed consent.

The questionnaire comprised one question on SRH, five questions on demographic characteristics, six questions on lifestyle items (e.g., wake-up time, bedtime, sleeping habits and BMI), five questions on miscellaneous health issues (e.g., feel tired, anorexia, dizziness, become irritable and cephalalgia), and 25 questions on food habits and attitudes towards food (e.g., frequency of eating breakfast, concerns about food, respect for food, watching TV while eating and talking about food with family). Both 4- and 5-point rating scales were used, with higher scores indicating more positive food habits. For example, questions regarding ‘respect for food’ utilised a 5-point rating scale (5 = high concern, 4 = middle concern, 3 = concern, 2 = little concern, 1 = no concern).

### 2.2. SRH Status

Global SRH measures generally include a question such as ‘How would you rate your overall health?’ and offer five response categories that range from excellent to poor (i.e., excellent, very good, good, fair and poor) [19,20,21]. In this study, we grouped scores of the SRH variable into two categories: the ‘healthy’ group included the responses ‘excellent’ and ‘very good’, whereas the ‘unhealthy’ group included the responses ‘good’, ‘fair’ and ‘poor’.

### 2.3. Statistical Analyses

The Chi-square test and Fisher’s exact test were used to evaluate differences in independent variables and SRH between the groups. *T*-tests were used to compare each variable between the self-rated healthy and unhealthy groups *p* values of < 0.05 were considered to indicate statistically significant differences.

Variables that were significant in the bivariate analysis were included in a multivariable logistic regression analysis with SRH as the dependent variable. The model was adjusted for gender and age, and a stepwise elimination method was employed for variable selection. Data were entered and analysed using SPSS for Windows, version 23.0 (IBM, New York, NY, USA).

Structural equation modelling was used to explore more complex food habit associations and alternative strategies for handling missing data. The following indices were used to assess model fit: goodness of fit index (GFI), adjusted goodness of fit index (AGFI), comparative fit index (CFI) and root-mean square error of approximation (RMSEA). Models were considered as having a good fit when values of GFI, AGFI and CFI were higher than 0.9, when the value of RMSEA was lower than 0.05, and values of AIC and CAIC were smaller. Model fit was investigated using Amos software, version 6.0 (IBM, New York, NY, USA).

### 2.4. Ethics Statement

Participants were well informed about the objectives and methods of this investigation. They voluntarily participated in the questionnaire survey without any compulsion, and with the right to freely withdraw from the study. Individual privacy was strictly protected during the entire investigation. This study was approved by the Kobe Women’s University Ethics Committee regarding Human Subjects.

## 3. Results

### 3.1. Sample Characteristics and Item-Specific Responses

Complete responses to the questionnaire items and sample characteristics are shown in Table 1. No statistically significant difference was observed between the two groups in terms of age, gender, grade, height, weight or BMI (*p* > 0.05, Table 1). In addition, there were no significant differences with regard to healthy vs. unhealthy or kind of school.

### 3.2. Relationships between SRH and Student’s Food Habits/Attitude

All variables related to lifestyle, food habit and attitude demonstrated statistically significant relationships (Table 2). The results suggested that positive lifestyle and food habits led to better SRH. In addition, the five variables related to miscellaneous health issues demonstrated a significant relationship with SRH (data not shown).

### 3.3. Multiple Logistic Regression Analysis

Results of the logistic regression showed that SRH was significantly positively associated with the frequency of eating breakfast and liking home meals (Table 3). In addition, these two items had a stronger relationship with SRH than the remaining lifestyle and miscellaneous health issues.

### 3.4. Structural Equation Modelling

Figure 1 displays the results of the structural equation model related to students’ food habits. The overall model fit was very good: CMIN = 0.083; CMIN/DF = 0.083; GFI = 1.000; AGIF = 1.000; NFI Delta = 0.999; CFI = 1.000; RMSEA = < 0.001; AIC = 10.083 and CAIC = 40.860. In the structural equation model, SRH was associated with the frequency of eating breakfast and liking home meals. We defined ‘liking home meals’ as the observable variable of the latent variable ‘attachment’. Likewise, we defined ‘frequency of eating breakfast’ as the observable variable of the latent variable ‘morning habits’.

## 4. Discussion

This research revealed that the frequency of eating breakfast and liking home meals was related to SRH in high school students. Each concept is discussed in detail in the following sections.

Table 3 suggests that ‘frequency of eating breakfast’ was associated with SRH. As mentioned in the Introduction section, SRH is related to the actual health status. The study revealed that male and female high school students who started smoking very early reported poorer SRH than students who had started later or had never smoked [22]. Based on these findings, it is suggested that SRH is the key factor for assessing the health of high school students.

Some research studies have suggested that the frequency of eating breakfast relates to the food habits and lifestyle patterns of adolescents. A decline in breakfast consumption has been reported during adolescence [23]. Furthermore, meal patterns, such as skipping breakfast, have been shown to be related to a group of less healthy lifestyle factors and food choices, leading to poorer nutrient intake [24]. Regular breakfast consumption has been associated with lower blood cholesterol and lower bodyweight [25]. Lastly, some studies have reported associations between skipping breakfast and increases in BMI or rates of metabolic syndrome [26,27]. Our results indicated that morning habits are a very important factor influencing the SRH of high school students, which may influence their future health.

In addition, our previous study clarified the association between food preferences and food habits of children, suggesting that morning habits have profound effects on concern and respect for food and food preferences [28]. In a study, it was concluded that breakfast contributes important nutrients to the diet of Australian children [29]. Our results indicated that morning habits may be related to health in adolescent students as well. Our results also indicated that sleeping habits were related to SRH (Table 2). Indeed, students with good sleeping habits (i.e., those who reported sleeping early and waking up early), may have sufficient time to eat breakfast every morning. Furthermore, people who had healthy lifestyles during childhood might more likely transfer healthy lifestyle patterns during adulthood to their own children in adulthood.

This study indicated that liking home meals was associated with SRH and morning habits. It has been shown that, during adolescence, the proportion of energy consumed at restaurants or fast-food establishments increases significantly, whereas that consumed at home decreases [30,31]. It is important to understand the factors that influence the preference of adolescents to eat at home rather than elsewhere. Watts et al. has described that targeting the home food environment is important to enable healthier food choices among overweight and obese adolescents [32]. In a study by Lee et al. [33] based on the Korea National Health and Nutrition Examination Survey, adults in the ‘non-home-made meal group’ tended to consume fried and grilled food, and had more one-dish meals, such as noodles and dumplings, compared with the ‘home-made meal group’. According to our study results, liking home meals is very important in terms of food choices, and may be connected with overall health. In addition, the results of the structural equation model suggested that ‘liking home meals’ was affected by the frequency of eating breakfast. In brief, this means that, adolescents who like home meals may have good morning habits and sufficient time to eat breakfast.

The results of the logistic regression and the structural equation model showed that ‘liking home meals’ was related to SRH. One research group examined the family social environment in childhood and SRH in young adulthood, and its results showed that enhanced family cohesion, particularly after family disruptions during childhood, promotes health in young adulthood [34]. Thus, families, and especially mothers, must cooperate with their children’s preferences regarding home meals. Mothers with higher SRH have a lower BMI [35]. Furthermore, the SRH in mothers is associated with their mothers’health. MacFarlane [36] has emphasized that parents should be encouraged to listen to and consider the food preferences of their adolescent children and provide supportive family mealtime environments and healthy foods at home. This argument is similar to our own. In addition, three of our previous studies have suggested that the attitudes of mothers towards food may have profound effects on children’s food preferences [37,38,39]. The attitudes of mothers towards their children’s diets may affect children’s food preferences not only in childhood, but also in adolescence. ‘Liking home meals’ was associated with concern of mothers about their children’s diet and health. This may be connected with Bowlby’s secure base theory [40,41]. A secure base is very important for a child’s healthy development. Adolescents who like home meals may have a good relationship with their parents at home. Thus, it is important that parents serve meals to their children so that children learn to like home meals. Above all, lifestyle factors, including mothers’ concern about their children’s diet, may be associated with the SRH of high school students.

In the current study, we assessed the relative importance of the lifestyles and food habits of students in their SRH via multiple regression analysis (Table 3). SRH was significantly related to the frequency of eating breakfast and liking home meals. In addition, this study analysed the associations between the SRH of students and healthy eating using structural equation modelling (Figure 1). The model showed that SRH was associated with the frequency of eating breakfast and liking home meals. Both the logistic regression analysis and the covariance structure analysis using structural equation modelling extracted the same two items—‘frequency of eating breakfast’ and ‘liking home meals’—from a pool of several items. This confirmed that the reproducibility and reliability of the results are sufficient. This finding suggested that the SRH of high school students is a key factor in assessing their health. In conclusion, the results of this study can benefit high school students, mothers and children.

### Study Limitations

Because we intentionally oversampled 11 schools, the sample analysed herein may not be nationally representative. Although it is very difficult to obtain nationally representative data, our results suggested the validity of this study in general. In addition, our findings are consistent with those of other researchers. As such, the results of this study may be useful for the future health of high school students and further life indices of mothers/children.

Our study had a cross-sectional design, and may only describe a temporary phenomenon. Nevertheless, this hypothesis needs to be tested in more depth. In future studies, we would like to analyse the results of interventions or cohort studies.

In addition, the study showed that SRH is associated with the frequency of eating breakfast, but it did not investigate the contents of breakfast. In a next study, we will investigate the contents of breakfast and their relationship with SRH.

## 5. Conclusions

SRH was strongly related to the frequency of eating breakfast and liking home meals among Japanese high school students. We demonstrated that these factors are very important because students reporting to frequently eat breakfast and like home meals had higher SRH. These students are more likely to have better health than other students. Therefore, students who like home meals during adolescence may also develop good morning habits and better SRH.

## Figures and Tables

**Figure 1 behavsci-07-00071-f001:**
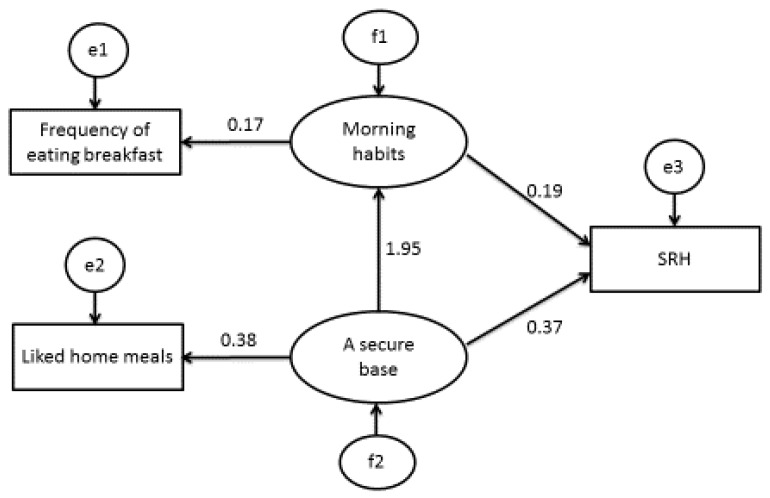
The structural equation model related to SRH and food habits (*n* = 1296). Note: Rectangles indicate observed variables and ovals indicate latent variables. Values of single arrowheads indicate standardised coefficients. All pathways are significant (*p* < 0.05). Self-rated health was associated with morning habits and a secure base.

**Table 1 behavsci-07-00071-t001:** Participant baseline characteristics classified by self-rated health (*n* = 1296).

	Variable	Self-Rated Health				
		Very Good to Excellent		Good to Poor		*p* Value
	All	1079	83.3%	217	16.7%	
Gender	Male	446	81.4%	102	18.6%	N.S *
	Female	633	84.6%	115	15.4%	N.S *
Grade	Grade 2	449	41.6%	97	44.7%	N.S *
	Grade 3	630	58.4%	120	55.3%	N.S *
	Height (m)	1043	162.5 ± 8.5	203	163.5 ± 8.1	N.S ^+^
All	Weight (kg)	971	54.8 ± 9.7	190	55.4 ± 10.6	N.S ^+^
	BMI (mg/kg^2^)	971	20.6 ± 2.8	190	20.6 ± 3.1	N.S ^+^
	Height (m)	441	169.8 ± 6.2	98	169.7 ± 5.7	N.S ^+^
Male	Weight (kg)	441	60.6 ± 9.7	96	60.6 ± 11.3	N.S ^+^
	BMI (mg/kg^2^)	439	21.0 ± 2.9	96	21.0 ± 3.5	N.S ^+^
	Height (m)	602	157.0 ± 5.3	105	157.7 ± 5.3	N.S ^+^
Female	Weight (kg)	530	50.1 ± 6.7	94	50.1 ± 6.5	N.S ^+^
	BMI (mg/kg^2^)	530	20.2 ± 2.7	94	20.1 ± 2.5	N.S ^+^
Bring Lunch Box						
	None	25	2.4%	6	2.9%	
	4 times per week or less	109	10.5%	32	15.4%	N.S *
	More than 5 times per week	908	87.1%	169	81.6%	

There were no significant differences between any subjects. Significance tested using chi-square * test and *t*-test ^+^.

**Table 2 behavsci-07-00071-t002:** Representative items affected by self-rated health (*n* = 1296).

		Self-Rated Health				
		Very Good to Excellent		Good to Poor		*p* Value *
Sleeping habits		N	%	N	%	
	Less 4 h	22	2.0%	13	6%	
	Over 4–5 h	152	14.1%	42	19.4%	
	Over 5–6 h	403	37.4%	72	33.2%	<0.001 ***
	Over 6–7 h	361	33.5%	70	32.3%	
	Over 7–8 h	135	12.5%	14	6.5%	
	Over 9 h	5	0.5%	6	2.8%	
Frequency of eating breakfast						
	None	80	7.4%	40	18.4%	
	1–2 times per week	50	4.6%	10	4.6%	<0.001 ***
	3–4 times per week	60	5.6%	17	7.8%	
	Over 5 times per week	886	82.3%	150	69.1%	
Frequency of eating breakfast with their family						
	None	428	40.2%	117	53.9%	
	1–2 times per week	118	11.1%	15	6.9%	<0.01 **
	3–4 times per week	94	8.8%	11	5.1%	
	Over 5 times per week	426	40.0%	74	34.1%	
Frequency of using convenience store						
	Every day	43	4.0%	15	6.9%	
	3–4 times per week	87	8.1%	30	13.8%	<0.01 **
	1–2 times per week	408	37.8%	84	38.7%	
	None	540	50.1%	88	40.6%	
Respect for food						
	None	2	0.2%	1	0.5%	
	Low respect	12	1.1%	7	3.3%	
	Medium respect	101	9.4%	34	15.8%	<0.000 ***
	High respect	357	33.2%	81	37.7%	
	Higher respect	602	56.1%	92	42.8%	
Concern about food						
	None	17	1.6%	6	2.8%	
	Low Concern	69	6.5%	28	13.2%	
	Medium Concern	204	19.1%	44	20.8%	<0.01 **
	High Concern	364	34.1%	73	34.4%	
	Higher Concern	414	38.8%	61	28.8%	
Liked home meal						
	Disliked	13	1.2%	5	2.3%	
	Low liking	55	5.1%	22	10.3%	
	Medium liking	193	18.0%	54	25.4%	<0.000 ***
	Liked a lot	318	29.7%	62	29.1%	
	Liked very much	491	45.9%	70	32.9%	

* Significance tested using Chi-square test and Fisher’s exact probability test. Note. ** *p* < 0.01, *** *p* < 0.001 by Fisher’s exact probability test.

**Table 3 behavsci-07-00071-t003:** Means and 95% confidence intervals of food habits by self-rated health in high school students after covariate adjustment.

	OR	(95% CI)	*p* Value
Frequency of eating breakfast	2.13	(1.07, 4.24)	0.031
Liked home meals	3.12	(1.27, 7.65)	0.013

OR, Odds ratio; CI, confidence interval. Multiple regression analysis used a stepwise method, adjusted for gender and age. Frequency of eating breakfast increased if many students ate breakfast more frequently.

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
