# Peer review of "Relationship between Self-Rated Health and Lifestyle and Food Habits in Japanese High School Students"

_behavsci, 2017, doi:10.3390/bs7040071_

Round 1

Reviewer 1 Report

Presented is a paper on the relationship of Self-Rated Health to lifestyle and eating habits in Japanese high school students.

The overall written presentation is satisfactory. English language requires moderate revision.

Introduction is too short and could be improved by more details, especially regarding the nature of SRH relationship with dietary habits.in general population.

In Discussionn you mention "An important study revealed that...". It should be explained why do you think this study is important?

Also, you stated "These findings suggest that SRH is the key factor for high school students’ health.". This statement should be further elaborated.

In Conclusions following statements are completely unclear and need to be rewritten: "These students should be taken to better health than others do not. Every meal is very important to adolescent that they like home meals."

Author Response

Answers to Reviewer 1: We appreciate the reviewer’s interest in and evaluation of our current study as well as his/her valuable suggestions. Reviewer 1’s comment 1. Presented is a paper on the relationship of Self-Rated Health to lifestyle and eating habits in Japanese high school students. The overall written presentation is satisfactory. English language requires moderate revision. In response to the comment, we have had the manuscript edited by an English editing service (Crimson Interactive Pvt. Ltd.). Reviewer 1’s comment 2. Introduction is too short and could be improved by more details, especially regarding the nature of SRH relationship with dietary habits in general population. According to the comment, we have added information to the introduction section. We have added more information in the introduction section with some references of the revised manuscript. Reviewer 1’s comment 3. In discussion you mention “An important study revealed that…”. It should be explained why do you think this study is important? According to your comment, we have changed the explanation in the Discussion section of our revised manuscript. As you said, the study was not very important. Reviewer 1’s comment 4. Also, you stated “These findings suggest that SRH is the key factor for high school students’ health.” This statement should be further elaborated. We appreciate the comments. We have added the explanation in the sentence of our revised manuscript. Reviewer 1’s comment 5. In Conclusions following statements are completely unclear and need to be rewritten: “These students should be taken to better health than others do not. Every meal is very important to adolescent that they like home meals.” Thank you for the comment. We have rewritten the statements conclusion in our revised manuscript.

Reviewer 2 Report

The title reads "Relationship between Japanese High School Students’ Self-Rated Health and . Lifestyle and Food Habits" . A better title would be "Relationship between Self-Rated Health and Lifestyle and Food Habits in Japanese High School Students" or "Relationship between Self-Rated Health and Lifestyle and Food Habits in High Schools in Japan".

The paper needs language corrections, and there are typos. To give you an example: The questionnaire consisted of SRH, five items related to 24 demographic characteristics, six lifestyle items.... The "," is needed.

This paper has assumed that constructs mean the same thing in Japan and other places. Please see that Self-Rated Health (SRH) mean different things in Japan compared to other locations: 

https://doi.org/10.1371/journal.pone.0183881

doi: 10.1016/j.annepidem.2015.11.006.
doi: 10.1007/s40615-017-0417-2.

doi: 10.1007/s40615-015-0154-3.

doi: 10.15171/ijer.2017.02.

There is also a difference in determinants of SRH across cultures and places:

 doi: 10.4103/2008-7802.164413. 

doi: 10.1186/2251-6581-13-36.
These papers also suggest that there is a need for local studies. Locally informed interventions are superior to universal interventions because of the role of context in shaping effects ...

So, the authors should discuss why local data is needed. Local data is needed because of the considerable cross-racial and cross-ethnic, and cross-country variations in meanings and correlates of SRH. 

A social epidemiological study may use the term gender not sex.

Do the results change if you look at various thresholds of SRH? What about predictors of excellent SRH? Or linear change in SRH (1 to 5)

If there are multiple ethnic groups, report the findings for each group. If there are multiple places (rural versus urban), report the results based on place. And report by gender as well...

Author Response

Answers to Reviewer 2: We appreciate the reviewer’s interest in and evaluation of our current study as well as his/her valuable suggestions. Reviewer 2’s comment 1. The title reads “Relationship between Japanese High School Students’ Self-Rated Health and Lifestyle and Food Habits”. A better title would be “Relationship between Self-Rated Health and Lifestyle and Food Habits in Japanese High School Students” or “Relationship between Self-Rated Health and Lifestyle and Food Habits in High Schools in Japan”. Thank you for your valid comment. According to your comment, we have chang “Relationship between Self-Rated Health and Lifestyle and Food Habits in Japanese High School Students” title. Reviewer 2’s comment 2. The paper needs language corrections, and there are typos. To give you an example: The questionnaire consisted of SRH, five items related to 24 demographic characteristics, six lifestyle itams… The “,” is need. In response to the comment, we have had the manuscript edited by an English editing service (Crimson Interactive Pvt. Ltd.). Reviewer 2’s comment 3. This paper has assumed that constructs mean the same thing in Japan and other places. Please see that Self-Rated Health (SRH) mean different things in Japan compared to other locations. There is also a difference in determinants of SRH across cultures and places. These papers also suggest that there is a need for local studies. Locally informed interventions are superior to universal interventions because of the role of context in shaping effects… So, the author should discuss why local data is needed. Local data is needed because of the considerable cross-racial and cross-ethnic, and cross-country variations in meanings and correlates of SRH. Thank you for your valid comment. According to the comment, we have added information to the introduction section with some references of the revised manuscript. Reviewer 2’s comment 4. A social epidemiological study may use the term gender not sex. Following the comments, we changed all the word to gender in the revised manuscript. Reviewer 2’s comment 5. Do the result change if you look at various thresholds of SRH? What about predictors of excellent SRH? Or liner change in SRH (1 to 5) This is a dichotomas logistic regression analysis. So the result did not change by various thresholds of SRH. This result suggested that the healthy SRH group had a higher frequency of “eating breakfast” and “liked home meals” to a greater extent. We predict that if the students do both them, they will be health in the future. Reviewer 2’s comment 6. If there are multiple ethnic groups, report the findings for each group. If there are multiple place (rural versus urban), report the results based on place. And report by gender as well… Thank you for your valid comment. According to your comment, we have added the place. There are some place, rural and urban. So, we have checked rural and urban that there was not significant differences healthy vs unhealthy and kind of schools. We added this sentence in the result section.
